# Associations between Perceptual Fatigue and Accuracy of Estimated Repetitions to Failure during Resistance Exercises

**DOI:** 10.3390/jfmk4030056

**Published:** 2019-08-09

**Authors:** Daniel A. Hackett, Victor S. Selvanayagam, Mark Halaki, Stephen P. Cobley

**Affiliations:** 1Exercise, Health and Performance Faculty Research Group, Faculty of Health Sciences, The University of Sydney, Lidcombe, NSW 2141, Australia; 2Centre for Sport and Exercise Sciences, University of Malaya, Kuala Lumpur 50603, Malaysia

**Keywords:** human performance, physical fatigue, fitness, perceived exertion, exercise training

## Abstract

The ability to accurately identify proximity to momentary failure during a set of resistance exercise might be important to maximise training adaptations. This study examined the association between perceptual fatigue and the accuracy of the estimated repetitions to failure (ERF). Twenty-seven males and eleven females performed sets of 10 repetitions at specific loads for the chest press and leg-press. Following the completion of 10 repetitions, participants rated their fatigue and ERF and then proceeded to concentric failure (actual repetitions to failure) to determine the ERF accuracy (i.e., error-ERF). Small correlations were found between perceptual fatigue and error-ERF for the chest-press (*r* = −0.26, *p* = 0.001) and the leg-press (*r* = −0.18, *p* = 0.013). For actual repetitions to failure and error-ERF, a strong correlation was found for the chest-press (*r* = 0.68, *p* < 0.001) and a very strong correlation was foundfor the leg-press (*r* = 0.73, *p* < 0.001). Moderate correlations were found between perceptual fatigue and actual repetitions to failure for the chest-press (*r* = −0.42, *p* < 0.001) and leg-press (*r* = −0.40, *p* < 0.001). Overall, findings suggest that the accuracy of the estimated repetitions to failure is more strongly associated with proximity to task repetition failure rather than subjective feelings of fatigue.

## 1. Introduction

Resistance training is a key form of strength and conditioning activity, whether utilised for clinical rehabilitation [1], health promotion [2] or in terms of athlete development and performance optimisation [3]. Within resistance training, performing sets of resistance exercises with close proximity to momentary failure appears to be beneficial for muscular hypertrophy in both novice [4,5] and advanced resistance trainers [6,7]. Briefly, momentary failure is defined as the set endpoint when a lifter attempts to complete the concentric portion of their current repetition but fails to do so [8]. Findings from Sampson et al. [9] suggests that performing resistance exercise to momentary failure is not critical for the development of muscle strength and hypertrophy. This is supported by data from a meta-analysis conducted by Davies et al. [10], which showed that similar gains in muscular strength can be achieved without performing sets of resistance exercise to momentary failure. However, resistance training performed to momentary failure for prolonged periods might result in overtraining and might increase the risk of musculoskeletal injuries [11,12]. Therefore, the ability to accurately estimate the number repetitions away from momentary failure seems valuable when practitioners or trainers are aiming to maximise resistance-training effects.

Resistance exercise intensity can be estimated from the rating of the perceived exertion (RPE) as the scale assesses subjective effort, strain, discomfort and fatigue. Although, there is evidence that the ability of RPE to discriminate momentary failure is poor [13,14]. The estimated repetitions to failure (ERF) scale [15,16] is a subjective method that can be used by health practitioners, strength and conditioning trainers, coaches and athletes, to monitor the proximity to momentary failure during sets of resistance exercise. The accuracy of using the ERF scale was investigated in a large cohort of male and females with varying degrees of resistance training experience [15]. Here, resistance training experience did not affect ERF accuracy, although males, relative to females, reported greater ERF accuracy during the resistance exercises [15]. These results might be related to anatomical-physiological differences in muscles that influence the central nervous system sensory perception capability of exertion, during exercise [17]. Other factors found to influence ERF include the type of exercise performed and proximity to momentary failure. More specifically in relation to the former, lower accuracy of ERF was found for the leg-press when compared to the chest-press [16]. In relation to the latter, ERF reporting, is more accurate when only a lower number of repetitions to failure (0–3 repetitions) can be performed when compared to a greater number (>5 repetitions) [15]. Therefore, accuracy of ERF was generally improved the closer the lifter was to momentary failure.

Exertion, which is a major feature of fatigue during exercise [18], is commonly assessed during exercise with the rating of perceived exertion (RPE) scale [19]. Exertion has previously been defined as the “degree of heaviness and strain experienced in physical work” [20], whereas, fatigue is defined as a disabling symptom where physical and cognitive function is limited by interactions between performance fatigability and perceived fatigability [21]. Previously, repetitions to momentary failure was found to be more strongly correlated with ERF, compared to RPE [16]. However, to the best of our knowledge no study to date has investigated the association between perceived fatigue and ERF. Since accuracy of ERF (error-ERF) is influenced by proximity to momentary failure, it would be expected that ERF accuracy would be strongly correlated with perceptual fatigue.

Thus, the purpose of this study was to investigate the associations between perceptual fatigue, actual repetitions to failure and error-ERF. The information gained from this study will provide knowledge on whether the ability to utilise exertional sensations during resistance training influences the accuracy of ERF.

## 2. Materials and Methods

### 2.1. Participants

Twenty-seven males (mean ± standard deviation of age = 25.6 ± 7.5 years; body mass = 76.5 ± 10.5 kg; height = 176.2 ± 8.2 cm) and eleven females (age = 26.1 ± 6.0 years; body mass = 59.6 ± 7.5 kg; height = 162.0 ± 6.7 cm) with recreational resistance training experience (males = 5.4 ± 6.3 years; females 4.6 ± 5.8 years) participated in this study. Each participant was risk assessed via the use of the American College of Sports Medicine pre-exercise screening questionnaire [22] and were deemed healthy to participate. An information statement explaining all procedures and study risks was provided to participants, alongside verbal explanations, prior to study participation. Verbal and written consent was provided by participants prior to study commencement. This study was approved by the University of Sydney Human Research Ethics Committee, project number 2013/924.

### 2.2. Study Design

Each participant was required to attend a familiarisation session and approximately one week later, an experimental session. Participants were instructed to maintain their normal diet during the days preceding the sessions; to consume their last meal at least two hours before exercise; and, to avoid using pre-workout supplements, nicotine, alcohol and caffeine on the day of the testing. Participants were further instructed to refrain from resistance training or any other strenuous type of exercise, 48 h prior to the sessions. During the familiarisation session, an estimated one-repetition maximum (1 RM) test was performed for the chest-press and the leg-press.

### 2.3. Estimated One-Repetition Maximum (1 RM)

Estimated 1 RM was performed for the pin loaded vertical chest-press machine (Maxim, Kidman Park, South Australia) and pin loaded horizontal leg-press machine (Kolossal, Sydney, New South Wales), prior to commencing the first experimental session. For each exercise, a standardised warm-up was performed (i.e., 1–2 sets of 8–10 repetitions with light-moderate loads), followed by the load adjustment, so that the participants could perform no more than 10 repetitions [23]. If >10 repetitions could be performed or failure was not reached prior to 10 repetitions, the load was increased and five minutes recovery was provided before the next RM attempt. The Brzycki 1 RM prediction equation [24] was used to estimate the 1 RM based on load and repetitions performed. The equation was mathematically expressed as: 1 RM = load/(1.0278–(0.0278 * number of repetitions)). Standard error of estimate of 1 RM using this equation for the chest-press was 1.67 and 3.00 kg, at 5 RM and 10 RM, respectively, and for the leg-press it was 13.74 and 20.41 kg, at 5 RM and 10 RM, respectively [23].

### 2.4. Familiarisation Session

After the 1 RM tests, participants were shown the ERF scale and the fatigue domain of the subjective exercise experiences scale. All participants received verbal instructions on how to use these scales. To assist participants to link exercise fatigue intensities and estimated capabilities with the full response of the respective range of the scales, a memory-anchor procedure was used. This involved asking participants to think of times when exercising where they reached levels of exertion equal to verbal cues, at the bottom and top of the scales. The familiarisation session involved participants following the same protocol, as used in the experimental session. Briefly, this involved participants performing five sets of 10 repetitions for the chest-press and leg-press, at a fixed percentage of one-repetition maximum (% 1 RM). At the completion of a set of 10 repetitions, participants were instructed to pause briefly and were asked to rate their fatigue responses, report their estimated repetitions to failure (using the ERF scale), and then continue to perform repetitions until momentary failure. A board with both the ERF and the fatigue domain of the subjective exercise experiences scales was positioned directly in front of the participants, during exercises, to allow generating immediate ratings. For the ERF ratings, participants were asked, “How many additional repetitions can you perform?” An ERF score of ‘10 or greater’ indicated an estimate of 10 or more repetitions could be completed, while a ‘0’ indicated that no additional repetitions could be completed (i.e., concentric failure). Participants were familiarised with both the subjective exercise experiences scale-fatigue and ERF scales, following 1 RM testing. For the fatigue domain of the subjective exercise experiences scale, participants were asked to provide a rating for how strongly they were fatigued, exhausted, tired and drained, along a 7-point Likert scale, ranging from 1 (not at all) to 7 (very much so).

### 2.5. Experimental Session

The experimental session began with each participant performing a warm-up that comprised of 8–10 repetitions, which were approximately 20% less than the loads used in the experimental sets, were performed for each exercise before the first set of the chest press and the leg press. After the warm-up and following the 1–2 min recovery, participants performed five sets of 10 repetitions at 70% of the predicted 1 RM for the chest press, and 80% of the predicted 1 RM was used for the leg-press. The rationale for using slightly different relative loads for these two exercises was to enable participants to perform a similar number of repetitions, to concentric failure [25]. During the exercises, participants were encouraged to complete each repetition through a full range of motion at a self-selected speed, however it was emphasised to keep the lifting speed constant. Following the completion of 10 repetitions, participants paused briefly (i.e., less than 10 s) at the end of the concentric phase, with full extension and locking out of the joints, and were asked to rate items related to fatigue (i.e., perceptual fatigue) and then ERF. Once participants reported their ERF, they then proceeded to perform repetitions until concentric failure (the actual number performed to failure—referred to as actual repetitions to failure,). There were 2–3 min for recovery between the sets of each exercise. The exercise order was randomised between sessions and there was approximately five minutes recovery between exercises.

Perceptual fatigue was assessed using the fatigue domain of the subjective exercise experiences scale–fatigue, which included the following items—fatigued, exhausted, tired and drained [26]. The subjective exercise experiences scale has been shown to have a high internal consistency across a variety of populations [27,28]. Participants were asked to provide a rating for how strongly they were fatigued, exhausted, tired and drained, along a 7-point Likert scale, ranging from 1 (not at all) to 7 (very much so). The subjective exercise experiences scale-fatigue score was calculated from the sum of the scores from the fatigued, exhausted, tired and drained items, with a maximum score of 28 (4 items multiplied by a maximum of 7 points). A board with the subjective exercise experiences scale-fatigue was positioned directly in front of the participants during exercises, to assist in generating immediate ratings.

### 2.6. Statistical Analysis

Statistical analyses were performed using the SPSS software package (Statistical Package for Social Sciences, version 24, SPSS Inc., Chicago, IL, USA). Data were presented as means ± standard deviation (SD) and the level of significance was set at *p* < 0.05. To examine the associations between perceptual fatigue, actual repetitions to failure and error-ERF, partial correlation analyses (adjusting for sex) were performed. For these analyses, data points from every set of all participants was used. The strength of correlations was qualitatively determined using the following criteria: trivial (*r* < 0.1), small (*r* > 0.1 to 0.3), moderate (*r* > 0.3 to 0.4), and strong (*r* > 0.5 to 0.7), very strong (*r* > 0.7 to 0.9), nearly perfect (*r* > 0.9 to 0.99) and perfect (*r* = 1.0) [29]. The error-ERF was calculated as the absolute difference between the ERF and the actual repetitions to failure for each set. Since the ERF scale was ‘‘0” to ‘‘10 or greater’’, any actual-repetitions-to failure value that was >10 was adjusted to 10. Perceptual fatigue, actual repetitions to failure and error-ERF between sets and sessions were analysed using a One-Factor Repeated Measures Analysis of Covariance (ANCOVA), with ‘sex’ as a covariate and ‘Time’ as the factor (i.e., sets). For significant ANCOVA results, Bonferroni corrections were applied. Effect size (ES) values were calculated as standardised differences in the means for any significant results. An ES of 0.2 was considered a small effect, 0.5 a moderate effect and 0.8 a large effect [30]. During the 5th set, 10 participants for the chest press and 3 participants for the leg press were unable to complete 10 repetitions. Therefore, due to this missing data, statistical analyses for perceptual responses, actual repetitions to failure and error-ERF was performed for sets 1−4.

## 3. Results

### 3.1. Perceptual Responses, Actual Repetitions to Failure and Error-ERF between Sets

There was a significant effect of set number on perceptual fatigue for the chest press at *p* < 0.05 [F(3, 147) = 19.64, *p* < 0.001]. Post-hoc comparisons revealed perceptual fatigue was lower for set 1 compared to sets 2 (*p* = 0.001, *d* = −0.83), 3 (*p* < 0.001, *d* = −0.121) and 4 (*p* < 0.001, *d* = −1.54), and for set 2 compared to set 4 (*p* = −0.009, *d* = −0.82), but there was no difference between any other sets (Table 1). This illustrated, as expected, that perceptual fatigue tended to increase from the initial to the latter sets, with a large ES being observed. There was a significant effect of the set number on the actual repetitions to failure for the chest press at *p* < 0.05 [F(3, 147) = 16.84, *p* < 0.001]. Post-hoc comparisons revealed actual repetitions to failure was greater for set 1 compared to sets 2 (*p* < 0.001, *d* = 0.96), 3 (*p* < 0.001, *d* = 1.08) and 4 (*p* < 0.001, *d* = 1.35), but was not different between any other sets. Therefore, participants had a greater number of actual repetitions to failure during the initial set relative to the latter sets, with a large ES being observed. There was a significant effect of the set number on the error-ERF for the chest press at *p* < 0.05 [F(3, 147) = 6.61, *p* < 0.001]. Post-hoc comparisons revealed that the error-ERF was greater for set 1, compared to sets 2 (*p* = 0.008, *d* = 0.67), 3 (*p* = 0.024, *d* = 0.56) and 4 (*p* = 0.001, *d* = 0.83), however, there was no difference in error-ERF between sets 2, 3 and 4. Therefore, ERF accuracy improved only after the initial set, with a moderate-to-large ES being observed.

There was a significant effect of set number on perceptual fatigue for the leg press at *p* < 0.05 [F(3, 147) = 15.03, *p* < 0.001]. Post-hoc comparisons revealed perceptual fatigue was lower for set 1 compared to sets 2 (*p* = 0.008, *d* = −0.65), 3 (*p* ≤ 0.001, *d* = −0.95) and 4 (*p* ≤ 0.001, *d* = −1.38) and for set 2 compared to sets 4 (*p* = 0.008, *d*=−0.80), but no difference was found between sets 3 and 4 (Table 1). This identified, as expected, that perceptual fatigue tended to increase from the initial to the latter sets, with a moderate-to-large ES being observed. There was a significant effect of the set number on the actual repetitions to failure for the leg press at *p* < 0.05 [F(3, 147) = 9.69, *p* < 0.001]. Post-hoc comparisons revealed actual repetitions to failure was greater for set 1 compared to sets 2 (*p* = 0.014, *d* = 0.59), 3 (*p* = 0.001, *d* = 0.79) and 4 (*p* < 0.001, *d* = 1.04), meaning that participants performed a greater number of repetitions during the initial set, with a moderate-to-large ES being observed. There was a significant effect of the set number on the error-ERF for the leg press at *p* < 0.05 [F(3, 147) = 4.87, *p* = 0.003]. Post-hoc comparisons revealed error-ERF was greater for set 1, compared to sets 3 (*p* = 0.003, *d* = 0.64) and 4 (*p* = 0.019, *d* = 0.76). Therefore, ERF accuracy improved only after the initial set, with a moderate ES being observed.

### 3.2. Associations between Perceptual Responses, Actual Repetitions to Failure and Error-ERF

Moderate correlations were found between perceptual fatigue and actual repetitions to failure for the chest-press (*r* = −0.42, *p* < 0.001) and the leg-press (*r* = −0.40, *p* < 0.001). Expectedly, this showed that perceptual fatigue was greater in participants when there were less actual repetitions to failure remaining during a set. For the actual repetitions to failure and error-ERF, a strong correlation was found for the chest-press (*r* = 0.68, *p* < 0.001) and a very strong correlation was found for the leg-press (*r* = 0.73, *p* < 0.001) (Figure 1). These results illustrated that when a participant had a lower number of actual repetitions to failure remaining in a set, the ERF accuracy was greater. Accompanying these findings, there were small correlations between perceptual fatigue and error-ERF for the chest-press (*r* = −0.26, *p* = 0.001) and the leg-press (*r* = −0.18, *p* = 0.013), suggesting a tendency for perceptual fatigue to increase when ERF accuracy was greater. However, this association was weaker compared to actual repetitions to failure and ERF accuracy.

## 4. Discussion

The purpose of this study was to investigate the associations between perceptual fatigue and ERF accuracy. There were small correlations found between perceptual fatigue and error-ERF. However, actual number of repetitions to failure showed stronger associations with both perceptual fatigue and error-ERF. This suggests that ERF accuracy is more strongly associated with the capacity to perform a task (i.e., proximity to repetition failure) rather than subjective feelings of fatigue. ERF accuracy was found to improve within sessions for both exercises, with the greatest error apparent in the initial sets of the chest and the leg press. During the initial sets, perceptual fatigue was lower and the actual number of repetitions to failure was greater. Therefore, ERF accuracy was possibly compromised due to the initial sets being less physically demanding.

Since the association between ERF accuracy and perceptual fatigue was weaker than that of the ERF accuracy and actual repetitions to failure, it seemed that factors related to exercise performance guides an individual’s ability to accurately estimate repetitions to failure. Previously, proximity to momentary failure was found to influence the accuracy of ERF [15]. Findings from the current study support this trend, with accuracy of estimated repetitions to failure improving when there was a lower number of repetitions to failure. The current study had participants perform successive sets using the same load with the same number of repetitions for the exercises. Therefore, an explanation for why the actual repetitions to failure influenced the ERF accuracy could be due to the participants recalling their performance in previous sets (e.g., overestimated or underestimated repetitions to failure) and then making subsequent estimate adjustments. Additionally, an important indicator of the ability to perform repetitions to failure is movement velocity [31], with velocity decreasing towards the end of sets and toward failure, where the slowest movement velocity is observed. Therefore, it is possible that participants utilise the movement velocity capability to guide ERF, and thereby account for the strong to very strong associations found between the ERF accuracy and the actual repetitions to failure [32].

Prolonged periods of resistance training performed to momentary failure might lead to overtraining and overuse injuries [11,12], and increase the risk of cardiovascular related adverse events [33]. Therefore, incorporation of ERF into resistance training might be a useful strategy for optimising performance or improving safety in specific clinical populations (e.g., elderly and people with hypertension). The accuracy of the ERF in the present study was considered reasonable, with the error-ERF across exercise sets being approximately 1 and 2–3 repetitions for the chest press and leg press, respectively. However, it could be argued that greater accuracy with ERF is needed to allow resistance training adaptations to be maximised. For instance, Giebetasing et al. [6] found that trainers performing sets to momentary failure experienced greater gains in muscular strength and hypertrophy, compared to trainers performing sets until an ERF of 1 repetition. Although, it should be noted that this previous study did not directly assess the accuracy of the trainers ability to ERF and the repetition tempo employed was super slow (10 s per repetition), which might have influenced the ERF accuracy.

Using ERF when programing for resistance exercise could assist with equating performances between trainers via selecting loads that allow them to be within a specific ERF range, following sets (e.g., 2–3 ERF). Such an approach would help to create more standardised individual exertion/fatigue responses following sets of resistance exercise, when performed using a specific % 1 RM for selected repetition ranges (e.g., 70% 1 RM for 8–10 repetitions). Another potential use for ERF during resistance training is to monitor individual responses throughout or between training sessions and determine how an athlete is progressing, and to indicate possible overtraining/overreaching states. As an example, if a trainer reported an ERF of 0, following the first set of 10 repetitions for the chest press using 70% 1 RM, while the previous week an ERF of 4 was reported for the same exercise prescription, this might indicate that further recovery time is required.

Although significant gains in muscular strength and hypertrophy are achieved with non-failure resistance training, it is recommended that the majority of sets are performed close to momentary failure, with the failure sets being used sparingly (e.g., the final set) [10]. Through the use of ERF, loads can be prescribed that can result in failure (or close to) during the final set, therefore, no decrease in load, and thus training volume, would be required to perform a specific number of repetitions. As an example, if 3 sets of 8 repetitions with 2 min recovery between sets were prescribed, selecting an initial load corresponding to an ERF of 3–4 would conceivably result in muscular failure being achieved by the 3rd set.

Notwithstanding key findings, the present observed results could be accounted for by several influential factors. For example, during exercise task completion, participants might have used their ERF as a goal, terminating the set once the target number of repetitions were achieved and thereby impacting the estimates of capability. Nevertheless, all participants were provided with consistent and equal encouragement in all sets to perform as many repetitions as they could until momentary failure. Resistance training experience is another factor that has previously been shown to influence ERF accuracy. For instance, Steele et al. [33] found greater ERF accuracy in trainers with more experience. Training experience can positively impact pain and fatigue tolerability, leading to performance of greater repetitions at specific relative loads (% 1 RM) [34]. Therefore, experience could moderate the association between perceptual fatigue and error-ERF, relative to a less trained individual. Within the present sample, participants were considered recreational and no information was gathered concerning their training history (e.g., training consistency, frequency, intensities, type of exercises, etc.). It can be assumed that participants had not trained in similar ways; therefore, such characteristics might have contributed toward the small to moderate associations between perceptual fatigue and error-ERF.

Since the resistance exercises were randomised, it is possible that the fatigue state of participants that performed the leg press, which is a more physically demanding exercise, first might have compromised the bench press performance. Additionally, the results might have been influenced by the nature of participant responses to assessments like perceptual fatigue, and its procedural application. While utilised for its high internal consistency across a variety of populations [27,28], the use of four similar anchor descriptions in close proximity might have led participants to consciously or unconsciously score all items in an (in) consistent manner, affecting the measurement accuracy. In the methodological approach, exact definitions of each perceptual fatigue item were not provided, leading to possible open interpretation and inconsistent use across participants. The findings from the current study are also limited as objective variables, such as velocity, muscle activity and other markers of fatigue, were not assessed. Therefore, future research is required to confirm the results of the present study with kinematics, electromyography and blood biochemical markers of fatigue. Finally, although 1 RM was estimated and not directly assessed, it is unlikely that any error would have led to large discrepancies between the relative training loads used by the participants. Furthermore, it is well-known that even at the same relative loads, individuals would perform a different number of repetitions to failure [25]. Therefore, two resistance trainers performing an exercise at the same relative loads might have different physiological and perceptual responses following a set of 10 repetitions. This further supports the argument that using an estimated 1 RM should not have confounded the present study findings.

## 5. Conclusions

Findings from this study suggest that perceptual fatigue might play a lesser role in ERF accuracy. Rather, it appears that factors related to proximity to repetition failure within resistance exercises might have greater association with ERF accuracy. If coaches, practitioners, or athletes are considering using ERF for training purposes, based on the current study findings it is possible that accuracy and reliability of ratings will vary between sessions. To assist with improving the ERF accuracy, it is recommended that the ERF scale is regularly utilised in training for familiarity and experience, and to improve their ability of utilising exertional sensations to increase the ERF accuracy. Additionally, periodic testing of ERF accuracy could be performed via calculation of the error-ERF. This might provide the trainer with useful feedback to determine forms of consistent ERF error, and provide adjustment accordingly to maximise benefits from resistance training without negative consequences. Since proximity to failure affects ERF accuracy, caution is warranted when interpreting ERF ratings for sets that are perceived as being less physically demanding.

## Figures and Tables

**Figure 1 jfmk-04-00056-f001:**
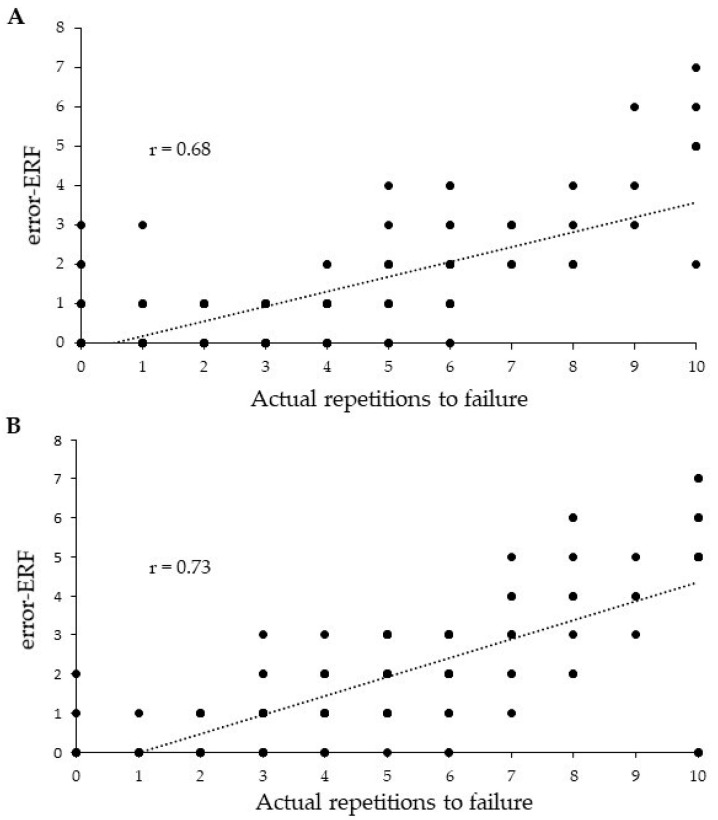
Scatter plot showing the association between error-estimated repetitions to failure (error-ERF) and actual repetitions to failure for the chest press (**A**) and leg press (**B**).

**Table 1 jfmk-04-00056-t001:** Perceptual fatigue responses, actual repetitions to failure and error-ERF during sets of the chest press and the leg press.

		Perceptual Fatigue	Actual Repetitions to Failure	Error-ERF
	Set	Mean ± SD (repetitions)	Mean ± SD (repetitions)	Mean ± SD (repetitions)
Chest press				
	1	17.24 ± 4.88 *	5.92 ± 2.70 *	1.97 ± 1.76 *
	2	20.84 ± 3.88 ^†^	3.66 ± 1.98	0.97 ± 1.17
	3	22.55 ± 3.84	3.08 ± 2.62	1.08 ± 1.42
	4	23.76 ± 3.49	2.47 ± 2.49	0.79 ± 0.99
	5	23.89 ± 3.82	2.14 ± 2.52	0.79 ± 1.26
Leg press				
	1	17.53 ± 5.27 **	7.76 ± 4.85 *	2.79 ± 2.21 **
	2	20.61 ± 4.15 ^†^	5.34 ± 3.15	2.03 ± 1.72
	3	21.76 ± 3.43	4.68 ± 2.58	1.55 ± 1.61
	4	23.66 ± 3.46	3.74 ± 2.47	1.34 ± 1.55
	5	24.71 ± 3.23	3.26 ± 2.68	1.37 ± 1.61

ERF—estimated repetitions to failure; error-ERF was calculated as the absolute difference between ERF and the actual repetitions to failure for each set. * Significantly different from sets 2, 3 and 4. ** Significantly different from sets 3 and 4. ^†^ Significantly different from set 4.

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
