# Peer review of "Associations between Perceptual Fatigue and Accuracy of Estimated Repetitions to Failure during Resistance Exercises"

_jfmk, 2019, doi:10.3390/jfmk4030056_

Round 1

Reviewer 1 Report

The authors investigate the associations between the perceptual fatigue, the Error of the Estimated Repetitions to Failure (E- ERF) and the Actual repetitions to failure. They found that the  association between the E-ERF with the actual repetitions was higher than with the perceptual fatigue.

These are original and interesting results that unfortunately do not add much to the theoretical understanding of the concept of failure and the relation among the studied variables. The well built experiment could be improved controlling the velocity of the task through a metrònom. One more repetition is always possible without the constrain of velocity. The authors should distinguish the concept of “task failure” from the concept of “fatigue-induced spontaneous termination point (FISTP) (see Hristovski & Balagué, 2010). It would be important to refer to it in the introduction and discussion sections. Authors should explain also why they didn’t study the dynamics of the variables under study (Balagué et al., 2015) and why they did not introduce objective variables (e.g., kinematic), together with the subjective ones (please see Vázquez, Hristovski & Balagué, 2016). Authors can improve the manuscript using the following references:

Balagué, N., Hristovski, R., García, S., Aguirre, C., Vázquez, P., Razon, S., Tenenbaum, G. (2015) Dynamics of Perceived Exertion in Constant Power Cycling: Time and Workload-dependent Thresholds. Research Quarterly for Sport and Exercise 86, 371-378

Balagué, N., Hristovski, R., Vainoras, A., Vazquez, P., Aragonés, D. (2014). Psychobiological integration during exercise. In: Davids, K., Hristovski, R., Araújo, D., Balagué, N., Button, C. & Passos, P. (Eds.). Complex Systems in Sport  (pp. 82-102). London: Routledge.

Hristovski, R. & Balagué, N. (2010). Fatigue-induced spontaneous termination point -Nonequilibrium phase transitions and critical behavior in quasi-isometric exertion. Human Movement Science, 29, 483–493.

Vázquez P, Hristovski R & Balagué N (2016). The Path to Exhaustion: Time-Variability Properties of Coordinative Variables during Continuous Exercise. Front. Physiol. 7:37. doi: 10.3389/fphys.2016.00037

Other minor problems of the manuscrit are related to:

Ln. 14. You are refering to “muscular failure”. Does failure occur only in the muscle? Please see for clarification: Hrsitovski & Balagué 2010, Balagué et al., 2014

Ln. 57. You refer to exertion or to perceived exertion?

Ln. 74. How did you calculated the sample size?

Ln. 118. The recovery time is not specified

Authors could enrich the discussion section reading: Capuccio, ML (Ed.) (2018). Handbook of Embodied Cognition and Sport Psychology. Cambridge: MIT Press

Author Response

We thank you for your constructive comments which have enabled us to improve the manuscript. Please find below a point-by-point response to all the comments raised. 

Reviewer Comment 1: “These are original and interesting results that unfortunately do not add much to the theoretical understanding of the concept of failure and the relation among the studied variables.”

Authors Response 1: We respectfully disagree with this synopsis because there has been no study to date that has explored perceived fatigue and its relationship to repetitions to failure, which does require inquiry to assist with more effective and individualised resistance exercise prescription.  Findings from the current study can be utilised to design a study that is mechanistic in nature and provide objective measures of fatigue as well as perceptual fatigue and whether there is any association with the accuracy of reporting repetitions to failure during resistance exercise.

Reviewer Comment 2: “The well-built experiment could be improved controlling the velocity of the task through a metrònome.”

Authors Response 2: Constraining a lifter to a specific velocity is not ideal since there are individual differences in the speed at which trainers habitually train. Furthermore, it is near to impossible to have a lifter perform all repetitions of set at a specific velocity since the lifting speed unintentionally continues to decline as repetition failure approaches (Izquierdo et al., 2006). Therefore, having a lifter perform repetitions at a self-selected speed allows the results to be more applicable and generalizable to the ‘real world’ for resistance trainers.

Izquierdo, M.; Gonzalez-Badillo, J. J.; Hakkinen, K.; Ibanez, J.; Kraemer, W. J.; Altadill, A., et al. Effect of loading on unintentional lifting velocity declines during single sets of repetitions to failure during upper and lower extremity muscle actions. Int J Sports Med. 2006, 27, 718-24.

Reviewer Comment 3: “One more repetition is always possible without the constrain of velocity.”

Author Response 3: To be constrained by velocity would mean that a ‘true’ repetition maximum would not be achieved. It is also important to note that changes in movement velocity is likely to influence the accuracy of estimating repetitions to failure i.e. a lifter will know that a limited number of repetitions are possible once the lifting speed has dramatically reduced. So this is another reason why we decided not to have participants perform repetitions at a specific lifting velocity.

Reviewer Comment 4: The authors should distinguish the concept of “task failure” from the concept of “fatigue-induced spontaneous termination point (FISTP) (see Hristovski & Balagué, 2010). It would be important to refer to it in the introduction and discussion sections.

Author Response 4: Yes we agree and have added related content to the Introduction (please see below). We do not feel it is necessary to include any further information related to the concepts of “task failure” and “fatigue-induced spontaneous termination point (FISTP)” within the Discussion section.

Task failure due to fatigue (i.e. fatigue-induced fatigue) is defined by the cessation of a bout of exercise and coincides with the inability to meet a previously defined performance criteria [8]. During resistance exercise, this can be observed by completing a full repetition (i.e. concentric followed by the eccentric phase and then ceasing the exercise). However, during exercise an individual usually can continue the exertion beyond the point when the performance criteria failure to meet. So during resistance exercise a lifter might be able to commence the concentric failure, however not be able to complete it. The term that is used in this circumstance is fatigue-induced spontaneous termination point (FISTP). Although, in the field of strength and conditioning this is generally referred to as ‘momentary failure’ and as such this, will be the term that is adopted throughout this paper [9]. “

Reviewer Comment 5: Authors should explain also why they didn’t study the dynamics of the variables under study (Balagué et al., 2015) and why they did not introduce objective variables (e.g., kinematic), together with the subjective ones (please see Vázquez, Hristovski & Balagué, 2016).

Author Comment 5: The dynamics of variables were not assessed due lack of resources at our university. This is also the same reason objective variables (e.g. kinematic) were not assessed. The following has been added to the Discussion.

“Finally, the findings from current study are also limited due to not assessing objective variables such as velocity, muscle activity, and other markers of fatigue. Therefore, future research is required to confirm the results of present study with kinematics, electromyography, and blood biochemical markers of fatigue.”

Reviewer Comment 6: Ln. 14. You are referring to “muscular failure”. Does failure occur only in the muscle? Please see for clarification: Hrsitovski & Balagué 2010, Balagué et al., 2014.

Author Comment 6: This sentence has now changed as shown below:

“The ability to accurately identify proximity to momentary failure during a set of resistance exercise may be important to maximise training adaptations.”

Reviewer Comment 7: Ln. 57. You refer to exertion or to perceived exertion?

Author Comment 7: This is exertion, not perceived exertion.

Reviewer Comment 8: Ln. 74. How did you calculated the sample size?

Author Comment 8: This was convenience sample and therefore sample size calculations were not performed.

Reviewer Comment 9: Ln. 118. The recovery time is not specified.

Author Comment 9: Yes it is “2-3 minutes recovery between sets of each exercise”.

Reviewer Comment 10: Authors could enrich the discussion section reading: Capuccio, ML (Ed.) (2018). Handbook of Embodied Cognition and Sport Psychology. Cambridge: MIT Press.

Author Comment 10: We believe that the Discussion sufficiently covers the main findings of this study and no further content is required.

Reviewer 2 Report

The authors have investigated associations between perceptual fatigue and accuracy of estimated repetitions to failure during the resistance exercises chest press and leg press. While the manuscript is in general well-written, I have some minor and major comments which might help the authors to improve the quality of their manuscript.  

Minor comments:

- Line 40 and 43: Please check and edit the brackets of the references. 

- Line 235: I think there is a one space too much between "during" and "initial sets". Please check and edit this sentence if necessary.

- Line 202/203: In Table 1 a bracket [ ")" ] is missing after "repetitions".

- Line 219: "ERF = estimated ..." should be changed into "ERF: estimated ...". 

Major comments:

- Line 94 and the following lines: Why did the authors estimated the 1RM using the Brzycki 1RM prediction equation rather than performing a "true" 1RM test? What are the reason(s) to use the equation provided by Brzycki instead of the equation provided by, for instance, Mayhew et al. (1992) or Wathan (1994)? Is the use of a estimated 1RM a confounding factor which might influence your findings? In my opinion adressing those questions in a revised version of the manuscript help the reader to better understand your research and findings.  

Mayhew, J. L., Ball, T. E., Arnold, M. D., & Bowen, J. C. (1992). Relative muscular endurance performance as a predictor ofbench press strength in college men and women. Journal of Applied Sport Science Research, 6, 200–206.

Wathen, D. (1994). Load assignment. In T. R. Baechle (Ed.), Essentials ofstrength training and conditioning (pp. 435–446). Champaign, IL: Human Kinetics.

- Line 170: Is there a special reason to use LSD (Least significant difference) instead of Bonferroni or Holm correction because LSD did not account for multiple comparison problem. A detailed discussion concerning multiple testing could be found in Bender & Lange (2001). This article may help the authors to justify their choice (e.g., exploratory study?). 

Bender & Lange (2001) - Adjusting for multiple testing—when and how? / https://www.ncbi.nlm.nih.gov/pubmed/11297884

- In my opinion the statistical analysis could be improved by calculating an appropriate effect sizes estimator (e.g., Cohen's d) because effect sizes are useful to determine and rate the "magnitude" of an observed effect. 

- With regard to the partial correlation analyses, it is not clear to me how the correlation coefficients are calculated in detail. It seems to me that an average value across all four sets was used to determine the correlation coefficients. If my assumption is true, why did the authors not calculate a partial correlation coefficient for every single set (e.g. first correlation coefficent for set 1, second correlation coefficient for set 2, third correlation coefficient for set 3, fourth correlation coefficient for set 4) or calculate within-subject correlation coefficients (see Dankel et al., 2017 or Bakdash & Marusich, 2017)?

Dankel et al. (2017) - Correlations Do Not Show Cause and Effect: Not Even for Changes in Muscle Size and Strength / https://www.ncbi.nlm.nih.gov/pubmed/28819744

Bakdash & Marusich (2017) - Repeated Measures Correlation  /  https://www.ncbi.nlm.nih.gov/pmc/articles/PMC5383908/

Author Response

We thank you for your constructive comments which have enabled us to improve the manuscript. Please find below a point-by-point response to all the comments raised. 

Reviewer Comment 1: Line 40 and 43: Please check and edit the brackets of the references. 

Author Comment 1: This has now been corrected.

Reviewer Comment 2: Line 235: I think there is a one space too much between "during" and "initial sets". Please check and edit this sentence if necessary.

Author Comment 2: This has now been corrected.

Reviewer Comment 3: Line 202/203: In Table 1 a bracket [ ")" ] is missing after "repetitions".

Author Comment 3: This has now been corrected.

Reviewer Comment 4: Line 219: "ERF = estimated ..." should be changed into "ERF: estimated ...". 

Author Comment 4: This has now been corrected.

Reviewer Comment 5: Line 94 and the following lines: Why did the authors estimated the 1RM using the Brzycki 1RM prediction equation rather than performing a "true" 1RM test?

Author Comment 5: The participants were recreationally trained and many had never performed a 1RM test before. Since in our laboratory have observed improvements of up to 20% in 1RM can be observed between an initial 1RM test and follow-up test a week later, it was decided that a predicted 1RM would be the most time efficient method to determine the load to use.

Reviewer Comment 6: What are the reason(s) to use the equation provided by Brzycki instead of the equation provided by, for instance, Mayhew et al. (1992) or Wathan (1994)?

Author Comment 6: Similarly put, we have used the Brzycki equation in past studies to estimate 1RM and have been satisfied with the results it produces in our participants.

Reviewer Comment 7: Is the use of an estimated 1RM a confounding factor which might influence your findings?

Author Comment 7: It is well known that even at the same relative loads individuals will perform a different number of repetitions to failure (Richens and Cleather, 2014). Since the task is related to repetitions maximum (RM) and ability to estimate RM, the actual relative load does not need to be exact between participants. I could perceive issues if the discrepancies were 50-90% 1RM but between 70-85% (which most likely everyone was training at) there would be no issues in terms of confounding the results.

Richens, B and Cleather, DJ. The relationship between the number of repetitions performed at given intensities is different in endurance and strength trained athletes. Biology of Sport 31: 157-161, 2014.

Reviewer Comment 8: In my opinion addressing those questions in a revised version of the manuscript help the reader to better understand your research and findings.  

Author Comment 8: The following has been added to the Discussion.

“Although 1RM was estimated and not directly assessed, it is unlikely that any error would have led to large discrepancies between the relative training loads used by participants. Furthermore, it is well known that even at the same relative loads, individuals will perform a different number of repetitions to failure [36]. Therefore, two resistance trainers performing an exercise at the same relative loads may have different physiological and perceptual responses following a set of 10 repetitions. This further supports the argument that using an estimated 1RM should not have confounded the present study findings.”

Reviewer Comment 9: Line 170: Is there a special reason to use LSD (Least significant difference) instead of Bonferroni or Holm correction because LSD did not account for multiple comparison problem. A detailed discussion concerning multiple testing could be found in Bender & Lange (2001). This article may help the authors to justify their choice (e.g., exploratory study?). 

Author Comment 9: We have changed the analysis from LSD to Bonferroni correction and made the required corrections to the results.

Reviewer Comment 10: In my opinion the statistical analysis could be improved by calculating an appropriate effect sizes estimator (e.g., Cohen's d) because effect sizes are useful to determine and rate the "magnitude" of an observed effect. 

Author Comment 10: We have added in effect sizes for any significant results found and included the following in the Statistical Analysis section:

“Bonferroni corrections were applied. Effect size (ES) values were calculated as standardised differences in the means for any significant results. An ES of 0.2 was considered a small effect, 0.5 a moderate effect and 0.8 a large effect [31].”

Reviewer Comment 11: With regard to the partial correlation analyses, it is not clear to me how the correlation coefficients are calculated in detail. It seems to me that an average value across all four sets was used to determine the correlation coefficients. 

Author Comment 11: Data from all sets of each participant was used for the partial correlation analyses and this information has now been added.

“For these analyses, data points from every set of all participants was used.”

Reviewer Comment 12: If my assumption is true, why did the authors not calculate a partial correlation coefficient for every single set (e.g. first correlation coefficent for set 1, second correlation coefficient for set 2, third correlation coefficient for set 3, fourth correlation coefficient for set 4) or calculate within-subject correlation coefficients (see Dankel et al., 2017 or Bakdash & Marusich, 2017)?

Author Comment 12: It is our opinion that combining all the data for the correlation coefficient analyses is sufficient for answering our research question.

Reviewer 3 Report

General Comments

This manuscript sought to investigate the associations between perceptual fatigue and estimated repetitions to failure. While this is an interesting topic that could be valuable to practitioners, a few suggestions and comments should be addressed to improve the work. 

Specific Comments

Abstract

Several p values were written as p=0.000, this should be changed to <0.001

Introduction

In the second paragraph the authors should elaborate on how there may be potential sex differences, especially since including male and female participants. For example, does muscle mass influence this? Absolute strength? Relative strength? Upper or lower-body exercises? 

Methods

What is the justification for sample size and was there an a priori power analysis performed?

How did the authors adjust for uneven groups within the statistical approach?

How do the authors define “recreational resistance training experience” although there was a range of years provided, what did the training constitute? 

Why did the design include an estimated 1RM procedure? The authors noted the SEE range in the methods but did notmention this in the discussion. This is a potential limitation resulting in uneven loads prescribed for the testing and must be mentioned within the discussion/limitation section. 

The authors provided a thorough explanation of the testing/verbal procedures and should be commended for their efforts- great job. 

What was a “moderate load” equivalent to for the experimental session? Both intensities (70 & 80%) generally can be depicted as moderate loads. Do the authors anticipate this may have contributed to fatigue? How much time was allowed between the warm-up and the testing repetitions?

When reporting the scale items following 10 reps, did the authors time how long it took for each participant to complete this procedure? Collecting these types of data often require the participant to think, thus it is hard to imagine all participants were able to provide the information in 5 seconds or less. Also, how did they “pause” were they in extension or flexion? Do the authors think that alternating from a consistent pace to a “pause and go” approach influence results?

Why were exercises randomized? What was the rationale for “approximately” 5-minutes recovery between exercises?

Do you think grouping the actual reps to failure (e.g. >10 reps) to 10 may have influenced the accuracy of this approach? How many people exceeded 10? 

Although I applaud the use of ANCOVA, was their other covariates that should be adjusted for (e.g. body weight)?

Results

Table 1 “repetitions” needs parentheses after the word.

Some of the p values have spacing between “p” and the value (e.g. p<0.001 & p = 0.000) these should be adjusted according to journal guidelines. 

Again, p values cannot equal 0.000

Figure 1 has a different font than the rest of the document, this should be adjusted to match the body of text. 

Discussion

Lines 275-279 the authors talk about velocity but did not require a consistent velocity across participants, do the authors speculate this could have contributed to the results?

How can this approach be used over the course of a training mesocycle? Would using a repetitions in reserve approach and gradually increasing toward 0 be a potential strategy for systematically increasing volume over the course of a training cycle? Please elaborate on the potential applications of your findings. 

Potential limitations: 

Different volumes for exercises- although a reference was cited and this is generally how exercise is completed, do you anticipate there may have been differences if the intensities were equal?

“pausing” and “reporting”- paused reps are more difficult to complete, with this reporting approach you incorporate both non-stop reps then paused back to non-stop, do you think this could have contributed?

Order of exercises- Do you expect there to be differences, or were there differences, for those who did leg press then bench versus bench then leg press? I would imagine that completing the leg press first would results in greater fatigue, which could compromise upper body performance. Although fatigue will accumulate with the bench press too, I am not certain it would be equal to that of a greater intensity bout during the leg press. 

Readings the authors should incorporate in the introduction and discussion:

Applications of the repetitions in reserve-based rating of perceived exertion for resistance training (Helms et al., 2016)

RPE vs percentage 1RM loading in periodized programs matched for sets and repetitions (Helms et al., 2018)

RPE and velocity relationships for the back squat, bench press, and deadlift (Helms et al., 2017)

Is repetition failure critical for the development of muscle hypertrophy and strength? (Sampson et al., 2016)

Author Response

We thank you for your constructive comments which have enabled us to improve the manuscript. Please find below a point-by-point response to all the comments raised. 

Reviewer Comment 1: Several p values were written as p=0.000, this should be changed to <0.001.

Author Comment 1: We have made these changes.

Reviewer Comment 2: In the second paragraph the authors should elaborate on how there may be potential sex differences, especially since including male and female participants. For example, does muscle mass influence this? Absolute strength? Relative strength? Upper or lower-body exercises? 

Author Comment 1: We have already mentioned that the accuracy of ERF is greater for males than females. We explained it may be the result of anatomical-physiological differences in muscle that influence central nervous system sensory perception capability of exertion during exercise. We do not feel that any further elaboration is required.

Reviewer Comment 3: What is the justification for sample size and was there an a priori power analysis performed?

Author Comment 1: It was a sample of convenience.

Reviewer Comment 4: How did the authors adjust for uneven groups within the statistical approach?

Author Comment 4: No statistical analyses were performed comparing males versus females so uneven groups was not an issue since there were combined. An ANCOVA was used and “sex” was included as a covariate.

Reviewer Comment 5: How do the authors define “recreational resistance training experience” although there was a range of years provided, what did the training constitute? 

Author Comment 5: The participants were not athletes and had a history of performing resistance exercise. No further information except years of resistance training experience was gathered.

Reviewer Comment 6: Why did the design include an estimated 1RM procedure? The authors noted the SEE range in the methods but did not mention this in the discussion. This is a potential limitation resulting in uneven loads prescribed for the testing and must be mentioned within the discussion/limitation section. 

Author Comment 6: The participants were recreationally trained and many had never performed a 1RM test before. Since in our laboratory have observed improvements of up to 20% in 1RM can be observed between an initial 1RM test and follow-up test a week later, it was decided that a predicted 1RM would be the most time efficient method to determine the load to use.

It is well known that even at the same relative loads individuals will perform a different number of repetitions to failure (Richens and Cleather, 2014). Since the task is related to repetitions maximum (RM) and ability to estimate RM, the actual relative load does not need to be exact between participants. I could perceive issues if the discrepancies were 50-90% 1RM but between 70-85% (which most likely everyone was training at) there would be no issues in terms of confounding the results.

Richens, B and Cleather, DJ. The relationship between the number of repetitions performed at given intensities is different in endurance and strength trained athletes. Biology of Sport 31: 157-161, 2014.

The following has been added to the Discussion.

“Although 1RM was estimated and not directly assessed, it is unlikely that any error would have led to large discrepancies between the relative training loads used by participants. Furthermore, it is well known that even at the same relative loads, individuals will perform a different number of repetitions to failure [36]. Therefore, two resistance trainers performing an exercise at the same relative loads may have different physiological and perceptual responses following a set of 10 repetitions. This further supports the argument that using an estimated 1RM should not have confounded the present study findings.”

Reviewer Comment 7: What was a “moderate load” equivalent to for the experimental session? Both intensities (70 & 80%) generally can be depicted as moderate loads.

Author Comment 7: The following was added.

“a warm-up that comprised of 8-10 repetitions of  approximately 20% less than the loads used in the experimental  sets were performed for each exercise before the first set of the bench press and squat.”

Reviewer Comment 8: Do the authors anticipate this may have contributed to fatigue?

Author Comment 8: No since the proximity to failure would have been large due to only 8-10 repetitions being performed with approximately 50-60% 1RM and there was 2-3 minutes recovery prior to the first set.

Reviewer Comment 9:  How much time was allowed between the warm-up and the testing repetitions?

Author Comment 9: The following was added.

 “After the warm-up and following 1-2 minutes recovery, participants performed five sets of 10 repetitions…”

Reviewer Comment 10:  When reporting the scale items following 10 reps, did the authors time how long it took for each participant to complete this procedure? Collecting these types of data often require the participant to think, thus it is hard to imagine all participants were able to provide the information in 5 seconds or less.

Author Comment 10: This was not timed, but we are very confident that is was less than 10 seconds. We have amended this information in the Methods. Participants would prepare to report these values during the last couple of repetitions prior to informing the researcher. As such, it was a fairly quick process and that is why we initially reported it took less than 5 seconds.

Reviewer Comment 11:  Also, how did they “pause” were they in extension or flexion? Do the authors think that alternating from a consistent pace to a “pause and go” approach influence results?

Author Comment 11: They paused at the end of the concentric phase, full extension and locking out the joints (this has been added to the Methods). No we do not believe that a pause following 10 repetitions would have significantly influenced the results.  

Reviewer Comment 12:  Why were exercises randomized? What was the rationale for “approximately” 5-minutes recovery between exercises?

Author Comment 12: This was done to eliminate any influence of the previous exercise. A 5-minute recovery is a standard amount of recovery time when performing maximal resistance exercise testing.

Reviewer Comment 13:  Do you think grouping the actual reps to failure (e.g. >10 reps) to 10 may have influenced the accuracy of this approach? How many people exceeded 10? 

Author Comment 13: 10 repetitions was exceeded by 3 participants only on the first set of the chest press, by 10 participants on the first set of the leg press, and by 3 participants on the second set of the leg press. No we do not believe this did influence the accuracy, since our findings already showed that accuracy was not good for set 1 compared to sets 2, 3, and 4.

Reviewer Comment 14: Although I applaud the use of ANCOVA, was their other covariates that should be adjusted for (e.g. body weight)?

Author Comment 14: We did explore other variables such as body weight, height, estimated 1RM, and these were found not to influence performance.

Reviewer Comment 15: Table 1 “repetitions” needs parentheses after the word.

Author Comment 15: This has been changed.

Reviewer Comment 16: Some of the p values have spacing between “p” and the value (e.g. p<0.001 & p = 0.000) these should be adjusted according to journal guidelines. 

Author Comment 16: This has been amended.

Reviewer Comment 17: Again, p values cannot equal 0.000

Author Comment 17: This has been amended.

Reviewer Comment 18: Figure 1 has a different font than the rest of the document, this should be adjusted to match the body of text. 

Author Comment 18: This has been amended.

Reviewer Comment 19: Lines 275-279 the authors talk about velocity but did not require a consistent velocity across participants, do the authors speculate this could have contributed to the results?

Author Comment 19: Constraining a lifter to a specific velocity is not ideal since there are individual differences in the speed at which trainers habitually train. Furthermore, it is near to impossible to have a lifter perform all repetitions of set at a specific velocity since the lifting speed unintentionally continues to decline as repetition failure approaches (Izquierdo et al., 2006). Therefore, having a lifter perform repetitions at a self-selected speed allows the results to be more applicable and generalizable to the ‘real world’ for resistance trainers.

Izquierdo, M.; Gonzalez-Badillo, J. J.; Hakkinen, K.; Ibanez, J.; Kraemer, W. J.; Altadill, A., et al. Effect of loading on unintentional lifting velocity declines during single sets of repetitions to failure during upper and lower extremity muscle actions. Int J Sports Med. 2006, 27, 718-24.

The following has been added to the Discussion.

“Finally, the findings from current study is also limited due to not assessing objective variables such as velocity, muscle activity, and other markers of fatigue. Therefore, future research is required to confirm the results of present study with kinematics, electromyography, and blood biochemical markers of fatigue.”

Reviewer Comment 20: How can this approach be used over the course of a training mesocycle? Would using a repetitions in reserve approach and gradually increasing toward 0 be a potential strategy for systematically increasing volume over the course of a training cycle? Please elaborate on the potential applications of your findings. 

Reviewer Comment 20: This may be an approach that could be used. However, we have added to the Discussion other approaches of how ERF could be using during resistance training (see below)

“Using ERF when programing for resistance exercise ERF could assist with equating performances between trainers via selecting loads that allow them to be within a specific ERF range following sets (e.g. 2-3 ERF). Such an approach would help to create more standardised individual exertion/fatigue responses following sets of resistance exercise when performed using a specific %1RM for selected repetition ranges (e.g. 70% 1RM for 8-10 repetitions). Another potential use for ERF during resistance training is to monitor individual responses throughout or between training sessions and determine how an athlete is progressing and indicate possible overtraining/overreaching states. As an example, if a trainer reported an ERF of 0 following the first set of 10 repetitions for bench press using 70% 1RM, while the previous week an ERF of 4 was reported for the same exercise prescription this may indicate further recovery time is required.”

“Although significant gains in muscular strength and hypertrophy are achieved with non-failure resistance training, it is recommended that the majority of sets are be performed close to failure, with failure sets used sparingly (.e.g. final set) [11]. Through the use of ERF, loads can be prescribed that can result in failure (or close to) during the final set, therefore no decrease is load, thus training volume, would be required to perform a specific number of repetitions. As an example, if 3 sets of 8 repetitions with 2 minutes recovery between sets were prescribed, selecting an initial load corresponding to an ERF of 3-4 would conceivably result in muscular failure being achieved at the 3rd set.”

Reviewer Comment 21: Different volumes for exercises- although a reference was cited and this is generally how exercise is completed, do you anticipate there may have been differences if the intensities were equal?

Author Comment 21: The following has been added to the Discussion.

“Although 1RM was estimated and not directly assessed, it is unlikely that any error would have led to large discrepancies between the relative training loads used by participants. Furthermore, it is well known that even at the same relative loads, individuals will perform a different number of repetitions to failure [11]. Therefore, two resistance trainers performing an exercise at the same relative loads may have different physiological and perceptual responses following a set of 10 repetitions. This further supports the argument that using an estimated 1RM should not have confounded the present study findings.”

Reviewer Comment 22: “pausing” and “reporting”- paused reps are more difficult to complete, with this reporting approach you incorporate both non-stop reps then paused back to non-stop, do you think this could have contributed?

Author Comment 22: Since the participants “locked out” their joints when reporting their values, this would be a rest and would have allowed maybe an extra repetition at most to be performed compared to non-stop repetitions. We do not think our approach affected the results.

Reviewer Comment 23: Order of exercises- Do you expect there to be differences, or were there differences, for those who did leg press then bench versus bench then leg press? I would imagine that completing the leg press first would results in greater fatigue, which could compromise upper body performance. Although fatigue will accumulate with the bench press too, I am not certain it would be equal to that of a greater intensity bout during the leg press. 

Author Comment 23: It is all relative, with some participants possibly finding some exercises most demanding then others. However, perceived fatigue increased as the sets increased  and since it was randomised, any effects from chest press on leg press performance is also accounted for, but as you have pointed out, possibly the performing  the leg press first might have altered the fatigue state prior to chest press performance. This has been acknowledged as a limitation as shown below:

 “Since the resistance exercises were randomised, it is possible that the fatigue state of participants that performed the leg press, which is a more physically demand exercise, first may have compromised bench press performance.”

Reviewer Comment 24: Readings the authors should incorporate in the introduction and discussion:

Applications of the repetitions in reserve-based rating of perceived exertion for resistance training (Helms et al., 2016)

RPE vs percentage 1RM loading in periodized programs matched for sets and repetitions (Helms et al., 2018)

RPE and velocity relationships for the back squat, bench press, and deadlift (Helms et al., 2017)

Is repetition failure critical for the development of muscle hypertrophy and strength? (Sampson et al., 2016)

Author Comment 24: We have added Helms et al., 2016 and Sampson et al. 2016 to the Introduction. We find the other references difficult to include in this manuscript without the flow to be disrupted and the word count being significantly over the limits.

We trust that the issues above have been addressed and clarified sufficiently.

Round 2

Reviewer 1 Report

The authors did some improvements to the manuscript but the main theoretical and methodological problems are still unsolved. Obviously, they did not revised adequately the recommended literature. 

Authors cannot mention the concept of fatigue-induced spontaneous termination point (FISTP) without citing it and, more importantly, understanding it.  Momentary failure" and "FISTP" are not equivalent and cannot replace each other. As the manuscript deals with endpoints in resistance training authors cannot use wrongly the terminology. In relation to the control of the velocity during the experiment, it is not enough saying that it is a limitation of the study. Velocity is a fundamental variable when assessing the number of repetitions and, in special when testing endpoints, I cannot recommend publication of the manuscript in the current form.

Author Response

We disagree with all of your comments and have decided to remove the information concerning FISTP. Your arguments are nonsensical and we recommend that you update your knowledge in the area of repetitions in reserve, perceived exertion, velocity-based training, and generally resistance training research.  

Our three papes below were all peer-reviewed and published in well-established high impact factor exercise science journals without any issues related to study design and in particular controlling of movement velocity. Please read these: 

Hackett, D., Cobley, S., Halaki, M. (2018). Estimation of repetitions to failure for monitoring resistance exercise intensity: Building a case for application. Journal of Strength and Conditioning Research, 32(5), 1352-1359.

Hackett, D., Cobley, S., Davies, T., Michael, S., Halaki, M. (2017). Accuracy in estimating repetitions to failure during resistance exercise. Journal of Strength and Conditioning Research, 31(8), 2162-2168.

Hackett, D., Johnson, N., Halaki, M., Chow, C. (2012). A novel scale to assess resistance-exercise effort. Journal of Sports Sciences, 30(13), 1405-1413.

Velocity should not be controlled when exploring the research question we investigated and the limitation we referred to is related to monitoring changes in velocity and NOT the controlling velocity. Usually it is instructed that participants perform the concentric phase at the maximal intended movement velocity or use a self-determined velocity. Please refer the studies below:

Ormsbee, M. J.; Carzoli, J. P.; Klemp, A.; Allman, B. R.; Zourdos, M. C.; Kim, J. S., et al. Efficacy of the Repetitions in Reserve-Based Rating of Perceived Exertion for the Bench Press in Experienced and Novice Benchers. J Strength Cond Res. 2019, 33, 337-45.

Garcia-Ramos, A.; Torrejon, A.; Feriche, B.; Morales-Artacho, A. J.; Perez-Castilla, A.; Padial, P., et al. Prediction of the Maximum Number of Repetitions and Repetitions in Reserve From Barbell Velocity. Int J Sports Physiol Perform. 2018, 13, 353-9.

Balsalobre-Fernandez, C.; Munoz-Lopez, M.; Marchante, D.; Garcia-Ramos, A. Repetitions in Reserve and Rate of Perceived Exertion Increase the Prediction Capabilities of the Load-Velocity Relationship. J Strength Cond Res. 2018.

Zourdos, M. C.; Klemp, A.; Dolan, C.; Quiles, J. M.; Schau, K. A.; Jo, E., et al. Novel Resistance Training-Specific Rating of Perceived Exertion Scale Measuring Repetitions in Reserve. J Strength Cond Res. 2016, 30, 267-75.

Reviewer 2 Report

The authors have adequately addressed my previous comments and have improved the quality of their manuscript. However, I have minor comments with regard to the reporting of statistics in the result section. 

- Line 256-332: I really appreciate that the authors have added effect sizes to the result section. However, I recommend that the authors should rearrange the order of reporting p-values and effect sizes. In the revised version of the manuscript the authors used following structure to report statistics: (effect size, p-value). This  should be changed into the commonly used structure: (p-value; effect size). For instance, the sentence (line 256/257).."Specific to the chest-press, perceptual fatigue was lower for set 1 compared to sets 2 (d=-0.83, p=0.001), 3 (d=-0.121, p<0.001) and 4 (d=-1.54, p<0.001)..." should be read as follows: "Specific to the chest-press, perceptual fatigue was lower for set 1 compared to sets 2 (p=0.001; d=-0.83), 3 (p<0.001; d=-0.121) and 4 (p<0.001; d =-1.54)..." Furthermore, I recommend to add the values of test statistic into the brackets (in order to follow APA guidelines). This means to report the statistics (e.g., t-test) as follows: (t-value (degrees of freedom), p-value; effect size). 

Author Response

Reviewer Comment: Line 256-332: I recommend that the authors should rearrange the order of reporting p-values and effect sizes. In the revised version of the manuscript the authors used following structure to report statistics: (effect size, p-value). This should be changed into the commonly used structure: (p-value; effect size).

Authors Response: We thank you for this suggestion and have amended the results as per your recommendation.

Reviewer Comment: I recommend to add the values of test statistic into the brackets (in order to follow APA guidelines). This means to report the statistics (e.g., t-test) as follows: (t-value (degrees of freedom), p-value; effect size).

Authors Response: We again thank you for this advice and have amended the results. Please see below:

“There was a significant effect of set number on perceptual fatigue for the chest press at the p<0.05 [F(3, 147) = 19.64, p<0.001]. Post hoc comparisons revealed perceptual fatigue was lower for set 1 compared to sets 2 (p=0.001, d=-0.83), 3 (p<0.001, d=-0.121) and 4 (p<0.001, d=-1.54) and for set 2 compared to sets 4 (p=-0.009, d=-0.82,), but there was no difference between any other sets (Table 1). This illustrated, as expected, that perceptual fatigue tended to increase from the initial to the latter sets, with large ES found. There was a significant effect of set number on actual repetitions to failure for the chest press at the p<0.05 [F(3, 147) = 16.84, p<0.001]. Post hoc comparisons revealed actual repetitions to failure was greater for set 1 compared to sets 2 (p<0.001, d=0.96), 3 (p<0.001, d=1.08) and 4 (p<0.001, d=1.35) but not different between any other sets. Therefore, participants had a greater number of actual repetitions to failure during initial sets relative to latter sets, with large ES found. There was a significant effect of set number on error-ERF for the chest press at the p<0.05 [F(3, 147) = 6.61, p<0.001]. Post hoc comparisons revealed error-ERF was greater for set 1 compared to sets 2 (p=0.008, d=0.67), 3 (p=0.024, d=0.56), and 4 (p=0.001, d=0.83), however there was no difference in error-ERF between sets 2, 3, and 4. So, ERF accuracy improved only after the initial set, with moderate to large ES found.”

“There was a significant effect of set number on perceptual fatigue for the leg press at the p<0.05 [F(3, 147) = 15.03, p<0.001]. Post hoc comparisons revealed perceptual fatigue was lower for set 1 compared to sets 2 (p=0.008, d=-0.65), 3 (p=<0.001, d=-0.95) and 4 (p=<0.001, d=-1.38) and for set 2 compared to sets 4 (p = 0.008, d=-0.80), but no difference was found between sets 3 and 4 (Table 1). This identifies, as expected, that perceptual fatigue tended to increase from the initial to the latter sets, with moderate to large ES found. There was a significant effect of set number on actual repetitions to failure for the leg press at the p<0.05 [F(3, 147) = 9.69, p<0.001]. Post hoc comparisons revealed actual repetitions to failure was greater for set 1 compared to sets 2 (p=0.014, d=0.59), 3 (p=0.001, d=0.79) and 4 (p<0.001, d=1.04), meaning that participants performed a greater number of repetitions during initial set, with moderate to large ES found. There was a significant effect of set number on error-ERF for the leg press at the p<0.05 [F(3, 147) = 4.87, p=0.003]. Post hoc comparisons revealed error-ERF was greater for set 1 compared to sets 3 (p=0.003, d=0.64), and 4 (p=0.019, d=0.76). Therefore, ERF accuracy improved only after the initial set, with moderate ES found.”

Reviewer 3 Report

The authors have made adequate changes and addressed all of the comments and concerns. Nice job elaborating on the methods and expanding the discussion. 

Author Response

Thank you for reviewing our manuscript and providing the recommendations to improve its quality.